# Polyethylene Glycol (PEG) Application Triggers Plant Dehydration but Does Not Accurately Simulate Drought

**DOI:** 10.3390/plants14010092

**Published:** 2024-12-31

**Authors:** Gulnar Kylyshbayeva, Nazira Bishimbayeva, Sativaldy Jatayev, Serik Eliby, Yuri Shavrukov

**Affiliations:** 1Faculty of Natural Sciences, Central Asian Innovation University, Shymkent 160000, Kazakhstan; kuntun-gulnar@mail.ru; 2Research Institute for Biology and Biotechnology Problems, Al-Farabi Kazakh National University, Almaty 050040, Kazakhstan; 3Faculty of Agronomy, S. Seifullin Kazakh AgroTechnical Research University, Astana 010011, Kazakhstan; s.jatayev@kazatu.edu.kz; 4Australian Centre for Plant Functional Genomics, University of Adelaide, Adelaide, SA 5064, Australia; selib1961@gmail.com; 5College of Science and Engineering, Biological Sciences, Flinders University, Adelaide, SA 5042, Australia

**Keywords:** containers with soil, dehydration, drought, field trial, gene expression, hydroponics, polyethylene glycol (PEG)

## Abstract

Polyethylene glycol (PEG), especially at high molecular weights, is highly soluble in water, and these solutions have reduced water potential. It is convenient to use PEG in hydroponics (liquid nutrient solution) for experiments with plants. However, some authors have been found to describe the application of PEG to plants incorrectly, such as drought, dehydration, osmotic, or water stresses, which can mislead readers. The presented opinion paper shows our arguments for a terminology in such experiments that is strictly limited to ‘PEG-induced’ or ‘simulated’ or ‘mimicked’ dehydration, and osmotic or water stresses, with the best option being ‘PEG-induced dehydration’. The most popular term, ‘drought’, is inappropriate to be used for hydroponics at all, with or without PEG. Traditionally, drought stress study was related to only plants in soil or other substrates mixed with soil. Based on 139 published papers, the examples presented in our opinion paper can demonstrate differences in gene expression between plants grown in containers with soil and under PEG-induced stress in hydroponics. Researchers can carry out any type of experiments suitable for the purposes of their study. However, clear and correct description of experiments and careful interpretation of the results are strongly required, especially with PEG, to avoid incorrect information. In all cases, at the final stage, results of experiments in controlled conditions have to be verified in field trials with naturally occurring drought.

## 1. Introduction

Readers cannot imagine the many thousands or even millions of papers in the literature that begin with a phrase along the lines of ‘Drought is the biggest threat …’ or similar. This indicates the extent to which drought is regarded as both an extremely important abiotic stress and one of the biggest challenges for living organisms, and plants especially, in conditions of global climate changes. Drought is defined as a single stretch of water deficit, lack, or cessation [1]. However, drought stress affecting crops and natural plant species has extremely complicated effects, where many factors, including the nature of plant species, developmental stage, geographic location, climate, soil, and many others, are involved [2]. Dehydration is an unavoidable component of drought and means loss of water from plant cells as a result of drought, heat, or salinity stress [3,4]. Dehydration is generally triggered by osmotic signals [5] and is an important part of plant reactions to drought. Osmotic stress occurs due to an imbalance of the osmotic gradient inside and outside of cells and usually requires rapid cellular responses to adjust and maintain a stable osmotic balance and adapt to challenging environmental conditions [6,7].

The study of natural drought is possible only in field trials, where plants grow in soil and rain-fed water is somehow stopped or limited. However, such natural environments are unpredictable due to fluctuations of climate and natural precipitation [8]. Therefore, for experiments studying plants under drought stress, artificially controlled and stable conditions with drying soil are much more suitable, particularly for reproducibility and repetition of experiments, including high-throughput phenotyping technologies [9]. Various bins, buckets, pots, and any other types of containers filled with soil are typically used in controlled greenhouses, chambers, cabinets, and even on laboratory benchtops to study the effects of drought on plants. However, in strong terms, such experiments must be classified as ‘artificially controlled drought’ compared to a ‘natural’ one. Nevertheless, this designation is very rarely used, perhaps because the description of ‘research field’ or ‘non-field’ conditions is obvious and clear in plant study.

In contrast, a dramatic difference can be made when artificial chemical compounds are used to absorb water and limit its availability to studied plants. Polyethylene glycol (PEG), usually with high molecular weight (6000 or 8000), remains the most popular reagent for such experiments, usually used at a concentration of 10–30% [10]. Liquid PEG is used as a supplement to growth solutions containing the nutrients required for plant growth, and despite quite variable protocols, descriptions, and applications, all experiments fall under the umbrella term of ‘hydroponics’. It was perfectly described on the web site ‘Use of PEG’ [11].

Only one paper has been found [12], where a PEG solution was applied directly to pots with transgenic *Arabidopsis thaliana* plants grown in Sunshine Mix #5 soil (Sungro, Chesapeake, VI, USA) mixed with vermiculite (3:1). The authors described it as ‘Chemical drought stress’ but it is hard to predict how PEG interacts with soil particles, organic and non-organic matter in soil. Nevertheless, it seems this method of PEG application is a very rare case [12]. Additionally, PEG can be added to agar-based tissue culture media in Petri dishes, which very convenient for model plant analyses or germinated seeds and young seedlings [4]. These methods of PEG treatments, together with soil and hydroponic strategies, were comprehensively reviewed with many examples [13]. However, the authors concluded that this method of using solid agar media with added PEG is very far from a direct extrapolation from laboratory models to field trials in dry conditions, but it remains very suitable for pre-screening of certain compounds in initial studies.

The aim of the presented paper was to justify our opinion that the application of PEG in any type of hydroponics is very often incorrectly described in publications as ‘drought’ or ‘PEG-drought’. Instead, the best option is to define it more accurately as ‘PEG-induced dehydration’. The assessment is provided for 139 published papers where both terms ‘PEG’ and ‘drought’ were used, and five cases were analysed in more detail where both drought in soil and PEG-induced dehydration were studied simultaneously.

## 2. Water Potential in Experiments with Soil and in Hydroponics

Experiments in containers with soil and in hydroponics represent two different types of study. Soil is a very complicated mix composed of inorganic and organic products, along with diverse communities of microorganisms existing in and around plant roots [14,15]. In contrast, hydroponics was designed for quick delivery of water and nutrients from the solution to plant roots [16,17].

Water potential Ψ_w_, or water available for plant uptake [4], can be reduced progressively in experiments with soil after watering is withheld, and this process is very dependent on soil type and its water capacity, volume, and overall density, which all affect the speed of water loss [18]. In some cases, water field capacity in soil or permanent wilting point was used in experiments with plants grown in containers with soil [19]. However, this required additional calculations to estimate water potential. There are some difficulties in precise measurement of Ψ_w_ in soil-based experiments, which is easier to do in hydroponics with the addition of PEG [20]. To compare different experiments with plants grown in soil and in solutions with PEG, it is important to arrange a similar level of water potential in leaves, Ψ_leaf_, which could be the most suitable indicator showing that plants have similar levels of water limitation in different conditions of growing (containers with soil or hydroponics). Two excellent examples of such a comparison are presented in greater detail in Section 4 below as special cases [21,22].

However, these two cases providing evidence of similar levels of measured water potential Ψ_leaf_ in plants grown in soil without watering and in hydroponics with PEG are very rare. The majority of all other studied publications only indicate the percentage of PEG added and include no study of plants in soil, since researchers are primarily interested in producing consistent and reproducible results. Therefore, concentrations of applied PEG and duration of treatments were determined empirically to reach an adequate level of stress [23] or were based on previous publications and initial tests [24,25].

The estimated reduction of water potential was present in experiments with the gradual addition of 10% PEG-6000 for plant treatments in hydroponics, where water (or osmotic) potential was reduced to reach levels of −1.48 MPa [24] or −1.50 MPa [26]. However, Ψ_w_ in solution is not the same as that within leaves and, therefore, such results cannot be directly compared with other data, particularly in experiments with soil.

## 3. Correct and Incorrect Description of Experiments with PEG Application to Plants

A portion of published papers was searched in the Scopus database using the terms ‘PEG’, ‘drought’, and ‘dehydration’ in the Title, Abstract, and Keywords. This yielded 143 published papers suitable for individual checking. Four papers were eliminated in the initial stage as studies involving tissue culture, calli, and embryonic cells, which are not suitable for the subject of the presented analysis. Results of the remaining 139 published papers are present in Table 1, with full details in Appendix A.

The largest number of papers (40%) described their experiments using PEG solution as a supplement in hydroponics as ‘PEG-treatment’ or sometimes as ‘PEG stress’. This is a very neutral description avoiding any question about the type of stress, which authors do not want to discuss as perhaps beyond the scope of their studies (Table 1).

The four most critical groups of publications described the conditions used to grow plants in hydroponics with PEG as follows: drought (26%), dehydration (18%), osmotic stress (7%), and water stress (9%). With some variation, about half of the published papers in each of these four groups described PEG application or supplements to hydroponics using the simple terms ‘drought, dehydration, osmotic and water stress’, respectively. This incorrect use of these terms for hydroponics accounts for more than ¼ of the studied papers in total (27% or 38 papers in Table 1). There is also a very interesting observation that each author’s team interpreted the application of PEG as they wished, depending on the focus of their research. For example, if the authors studied drought, their descriptions and discussions of PEG treatment experiments were only related to drought using exclusively this term. Similarly, if the topics of papers concerned dehydration, osmotic or water stress, only these terms were used, respectively.

The biggest problem we see for these authors is that the application of PEG is non-direct and artificially causes effects in plants, which differ from the traditionally described phenomenon of drought, dehydration, osmotic or water stress. Therefore, authors from the latter groups showed much better descriptions, using the terms ‘PEG-induced’, ‘simulated’, ‘mediated’, or ‘mimicked’ (33% or 46 papers in Table 1, indicated in bold). These authors emphasized that PEG application causes indirect effects and needs to be described with additional words like ‘induced, simulated, mediated’, and so on. Such a decision is the best option since it clearly indicates correct use of the scientific terms for the description of their experiments with PEG treatment (Table 1).

## 4. Analysis of Special Cases with Experiments Both in Soil/Substrate and Hydroponics

Below, five papers were selected as examples where authors studied plants grown in both soil/substrate and hydroponics with PEG simultaneously. For these cases we compared the published results for gene expression analyses when it was possible. All presented cases are very different, and it was particularly of interest to show studies and results with very diverse plant species or different stages of plants and the specific experimental designs. Our attention was especially directed to the Materials and Methods section and how the authors described the conditions they used to grow plants in soil or substrates and PEG-treated hydroponics and what kind of results were reported.

### 4.1. Case 1: Cotton Plants in Hydroponics with Sand and PEG Treatment

The first selected paper was a study of the comparison of cotton plants grown “under stress of PEG-drought compared with Soil-drought” [21]. Despite the authors’ statement, there was no soil but only sand as the substrate with added growth solution. So, in fact, it was a study of a “culture matrix” with or without the addition of PEG solution. Nevertheless, the authors in this study were particularly focused on the identical water potential in leaves, Ψ_leaf_, as described in Section 2 above [21].

Three-leaf stage young plants of the upland cotton accession Zhong H177 were grown in sand as the culture substrate. For PEG treatment, growth solution with 5% PEG was used for growing plants for 6 h. For dehydration without PEG, growth solution was withdrawn for three days, and plants were exposed to slowly drying sand. The very important step was to choose conditions of treatments and to make identical water potential in leaves of plants in both treatments, Ψ_leaf_ = −2.07 MPa. It made possible the near-perfect comparison of plants from different experiments but with the same water potential [21].

In the results, the authors concluded that the physiological and biochemical characteristics in the studied cotton plants were similar in both treatments, with and without PEG. However, significant differences were found in the expression of seven genes related to photosynthesis, which were up-regulated during slow dehydration without PEG, but the expression of the same genes was steady or down-regulated in the presence of 5% PEG. The similar increased expression levels were reported for eight glucose metabolite-related genes, three sugar transporter genes, and some other genes in slowly drying sand without PEG compared to PEG treatment but with the same level of water potential in leaves [21]. Therefore, PEG application and slowly drying sand had different effects on plants with the same water potential.

### 4.2. Case 2: Medicago Truncatula Plants in Substrate and PEG Treatment

In the second case, seedlings of *Medicago truncatula* were grown for 11 weeks in a mix of perlite–vermiculite (1:3), irrigated with growth solutions. The substrate mix used is artificial and commonly used in laboratory experiments where absorption and retention of moisture for a certain time is required, but this is still different from real soil. In this experiment, no genes were analysed but only physiological and biochemical parameters [22].

After 11 weeks of growth, *M. truncatula* plants were treated for 7 days either with a 2/3 reduction of irrigation in the substrate mix, indicated as ‘No-W (no watering)’, or with the addition of 25% PEG-6000 solution. Leaf water potential was measured and confirmed as identical in both treatments, Ψ_leaf_ = −1.68 MPa, making equal dehydration pressure, corresponding to a moderate water-deficit level [22].

Based on the comparative analysis, the authors concluded that some physiological parameters, including stomatal conductance and leaf transpiration, were the same in both the No-W and the PEG studies. However, accumulation and content of water were significantly higher in roots and lower in the leaves of *M. truncatula* plants grown in PEG compared to No-W treatment. Similar differences were found for carbohydrate accumulation in leaves, significantly higher for sucrose content and a remarkable ‘zero’ accumulation of starch in leaves after PEG treatment compared to exactly opposite trends in No-W plants. The authors reported on many other studied compounds and traits showing big differences between No-W and PEG treatments, and they finally summarised that PEG is “a false mimetic of drought”. It is important to emphasize that leaf water potential was identical in the presented experiments [22].

### 4.3. Case 3: Wild Barley Plants in Soil, Leaf Dehydration, and PEG Treatment

The next case was related to gene analysis, but water potential was not measured or reported in this study. Young barley plants (four-leaf stage) were studied in several conditions, including the following: (1) gradual drought in pots with soil (local soil, nutrient soil, and vermiculite in a ratio of 7:2:1); (2) rapid dehydration of detached leaves on a bench at room temperature (called ‘dehydration shock’); and (3) 20% PEG 6000 supplied to 1/8 liquid MS medium (hydroponics) [44]. Relative expression of the *HvLRX* (light-responsive X), described by the authors as a new dehydration- and light-responsive gene, was analysed in two drought-tolerant and sensitive accessions.

Under drought stress, the *HvLRX* expression gradually increased during 5 and 7 days of drought, reaching about three units in leaves of both accessions. In contrast, under rapid dehydration, time points and magnitudes of *HvLRX* expression were sharply up-regulated over 1 and 5 h to about 17–22 units. However, results of *HvLRX* expression in hydroponics with PEG in leaves of the same barley accessions was dissimilar to both rapid dehydration and gradual drought, instead fluctuating between about 2.2 and 3.2 units with the maximum after 8–12 h of PEG treatment [44]. The results of this study indicated that PEG treatment showed the induction of plant reaction to stress; however, the simulation of real drought or dehydration conditions in this case was very limited.

### 4.4. Case 4: Breeding Lines of Maize Grown in Soil, Dehydrated Plants, and PEG Treatment

Young maize plants (three-leaf stage) were studied in the following conditions: (1) gradual drought in plastic containers filled with a mixture of nutrient soil and vermiculite (1:1); (2) rapid dehydration treatment with plants pulled from pots and air-dried on paper; and (3) 20% PEG treatment. Plants grown in vermiculite were rinsed and immersed in Hoagland solution containing PEG. Relative expressions of NAC transcription factor genes, *ZmNAC87* and *ZmJUB1*, were analysed in maize inbred lines Lv28 and H082183 [45].

In an experiment with moderate and severe drought (several days of treatment), expressions of both genes were mixed in both roots and shoots in the studied breeding lines, reaching maximums of 8–12 unit levels for gene *ZmNAC87* and 6–8 unit levels for *ZmJUB1*. Very different results were reported for rapid dehydration, where expression of *ZmNAC87* was highest in shoots (30 units, 3 h) and in roots (35–55 units, 3–24 h). The expression of *ZmJUB1* was recorded at levels 5–35 in shoots and 1–5 units in roots. Importantly, some similarity was found in *ZmJUB1* expression during PEG treatment, which varied between 3–15 units in shoots and 1–5 in roots. It was particularly clear for breeding line Lv28 but more variable in breeding line H082183. In contrast, expressions of *ZmNAC87* in PEG-treated plants were very low and quite different from those in the dehydration experiment [45]. The presented results indicated relative similarity between experiments with rapid dehydration and PEG treatment in hydroponics for *ZmJUB1* expression but not for the *ZmNAC87* gene. However, it is important to note that water potential was not measured in this study, and, therefore, we cannot exclude the possibility that the differences observed between treatments may be more related to variations in water potential than to the treatments themselves.

### 4.5. Case 5: Soybean Plants in Soil and PEG Treatment at Different Stages

For the gradual drought experiment, water was withheld from young soybean plants (25 days after seed sowing, V4 stage) grown in pots with a mixture of soil and vermiculite (2:1). PEG treatment (20%) was applied to other soybean plants in hydroponics at the V1 stage. Expression of 15 genes from the *GmLAX* (like Auxin) family genes was measured in one soybean cultivar, Tianlong 1 [46].

All studied genes were down-regulated in response to either drought or PEG treatment. However, a big difference was observed between the control and drought-exposed plants whereas the difference was much smaller in the PEG-supplement experiment. The transcript level of *GmLAX3* was very similar in both treatments, but other genes, such as *GmLAX2* and *GmLAX13*, showed significant differences in their expressions under drought compared to experiments with PEG treatment [46]. In addition to the absence of water potential measurement, a mixed conclusion can be made for the analysis of the present paper since it remains unclear what role the different stages of plant development in the studied soybean—V1 stage in PEG treatment and V4 stage in drought—played in the final analysis. Nevertheless, there were big differences between drought and PEG treatments, and the authors attempted to separate the results from the ‘drought in soil’ experiment and ‘PEG-treatment in hydroponics’.

## 5. Conclusions

Researchers are permitted to use any type of experiments suitable for the purposes of their study. However, clear and correct description is required to avoid inferring incorrect information and potentially misleading their readers. It is recommended to use the terms gradual or slowly ‘induced drought’ or ‘simulated drought’ in containers with drying soil. At the same time, the best option for experiments with hydroponics supplemented with PEG solutions are the terms ‘PEG-induced dehydration’ or ‘PEG induced dehydration simulating drought’. Alternatively, ‘PEG-treatment’ experiments can be used and described properly. However, the authors must be especially careful in the interpretation of their results since experiments with PEG cannot be considered as true ‘drought’ studies. In all cases, results of experiments in controlled conditions must ultimately be verified in field trials with naturally occurring drought.

## Figures and Tables

**Table 1 plants-14-00092-t001:** Analysis overview of 139 published papers containing PEG application to plants in hydroponics and author descriptions of their experiments as either PEG treatment, drought, dehydration, osmotic or water stress. Special terms are indicated in bold.

Description	Sub-Description	No. of Papers	%	Description in Abstracts or in the Text of Papers	Ref.
PEG treatment		55	40	PEG treatment; PEG stress	[27,28]
Drought	Drought	17	26	Drought stress; drought (PEG treatment)	[29,30]
	PEG-induced drought	19		PEG-**induced** drought; PEG treatments to **simulate** drought stress; drought stress **induced** by PEG	[31,32]
Dehydration	Dehydration	12	18	Dehydration stress; dehydration tolerance in response to PEG treatment	[33,34]
	PEG-induced dehydration	13		PEG-**induced** dehydration; PEG-**simulated** dehydration; PEG-**mimic** dehydration	[35,36]
Osmotic stress	Osmotic stress	4	7	Osmotic stress; PEG osmotic stress	[37,38]
	PEG-induced osmotic stress	6		PEG-**induced** osmotic stress; PEG-**simulated** osmotic stress	[39,40]
Water-stress	Water-stress	5	9	Water stress; water-deficit stress	[41,42]
	PEG-induced water- stress	8		PEG-**induced** water deficiency; PEG-**mediated** water stress; PEG-**simulated** water deficit; water stress **induced** by PEG; water stress **mimicked** by PEG	[25,43]
Total		139	100	PEG supplement (application) in hydroponics	

## Data Availability

The original contributions presented in this study are included in the opinion paper and in the Appendix A. Further inquiries can be directed to the corresponding author.

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
