# Peer review of "Polyethylene Glycol (PEG) Application Triggers Plant Dehydration but Does Not Accurately Simulate Drought"

_plants, 2024, doi:10.3390/plants14010092_

Round 1

Reviewer 1 Report

Comments and Suggestions for Authors

Title: Polyethylene Glycol (PEG) Application Simulates Plant Dehydration, but It Is not Accurately Drought

Authors: Gulnar Kylyshbayeva , Nazira Bishimbayeva , Sativaldy Jatayev , Serik Eliby , Yuri Shavrukov

PEG is a non-toxic polymer that alters osmotic potential and induces dehydration in plant materials, making it widely used to simulate or induce drought stress in a controlled manner. Depending on the medium (such as MS medium or hydroponics) and the study's objectives, PEG can be applied as an inducer of water stress, dehydration, or osmotic stress. This is why different citations are found in literature.

Studies using PEG are commonly conducted with model plants, such as Arabidopsis thaliana, to induce drought stress and assess the pathways involved in stress response mechanisms. These findings can then be extrapolated and validated in soil-based experiments, for example. Additionally, the use of PEG in hydroponic or culture media, such as MS, offers a more direct, visible, and rapid way to study root development under stress conditions compared to soil, which can sometimes hinder root growth.

The manuscript presents case studies on various species, comparing the expression of stress response genes in soil and hydroponic conditions using PEG. Generally, the studies cited assess the response of specific genes to various stress conditions. For instance, Ding et al. (2022) evaluated the expression of NAC genes and found that the overexpression of certain genes enhances drought resistance in Arabidopsis thaliana. However, the aim of these studies was not to directly compare the treatments that induced stress, which is why they did not analyze the water potential between treatments. As a result, the differences observed between treatments, such as drought stress in soil and PEG use, may be more related to variations in water potential rather than the treatments themselves. To better address the authors' proposal, it would be more relevant to consult studies that standardize water potential across different treatments. When comparing gradual drought in soil, dehydration of detached leaves, and MS medium supplemented with PEG without standardizing water potential, the variations in gene expression likely reflect the intensity of stress in each treatment, rather than the type of treatment itself.

I expected the manuscript to offer a concise and informative opinion, along with a clear methodology on the use of PEG to mimic drought stress. However, the topic was addressed in a superficial manner, lacking a solid scientific foundation. Regarding the only figure in the manuscript, the caption lacks sufficient description of the images (e.g., species, type of cultivation), and the figure is not adequately discussed in the text.

Comments on the Quality of English Language
Can be improved

Author Response

Please see our responses in the attached file

Reviewer 2 Report

Comments and Suggestions for Authors

The article addresses an intriguing topic regarding whether treatment with PEG (polyethylene glycol) is synonymous with terms like drought, osmotic stress, dehydration, or water stress. Personally, I have some doubts about this methodology and its actual implications, both physiologically and molecularly, making this a fascinating and controversial subject worth exploring.

In the introduction, the authors provide appropriate definitions to understand the subtle differences between drought and dehydration. However, I would like to see more detail about the action of PEG in this context to better frame their perspective. The bibliographic research conducted, along with Tables 1 and S1, is both adequate and necessary.

The special cases analyzed are well chosen, though it might also be interesting to include examples mentioned earlier, such as the application of PEG directly in soil or in hydroponic systems, to allow comparisons and provide a more comprehensive spectrum of analysis.

Author Response

(The authors gave the same response as above.)

Reviewer 3 Report

Comments and Suggestions for Authors

Major points :

The difference between PEG treatment and drought is an interesting question to raise and I agree that these 2 stresses are not exactly the same. A PEG treatment doesn’t take into account the complexity of the soil and usually not gradually applicated, as natural drought stress. However, PEG generates a decreased water potential which is exactly the consequence of a drought stress. In this way, it mimics the effect of this natural stress. For example both soil drying and PEG lead the cells to experience cytorrhysis (shrinking of both plasma membrane and cell wall) whereas a NaCl treatment only triggers a plasmolysis (shrinking of plasma membrane only). This is well defined in the review Haswell and Verslues (2015), J Gen Physiol, 145:389-394. The authors should look at the work of Verslues and coll who did a lot on drought and osmotic stress (Verslues et al., 2006, The plant Journal; 45: 523-539).

Furthermore, there is also a difference between drought and dehydration. Dehydration can be the consequence of a drought stress as well as a heat stress. In this latter stress, the plant can experience a dehydration although the soil is relatively wet.

Although this literature review is interesting, the resulting presentation of this article is rather light. It is mostly descriptive and not enough conclusive. For example, the section 2 is very short and the associated figure doesn’t bring any information at all. Moreover, the 3 “in-depth” examples are not very conclusive since it concerns different plants at different developmental stages with analysis of some genes for which we don’t know the link to drought stress (LRX gene especially).

According to these remarks, I propose to the authors to state a clear definition of drought, dehydration, osmotic stress and try to see difference with PEG experiments from a time perspective. I mean that maybe the real difference is that natural drought is a gradual stress whereas PEG application is rather a shock stress.

Minor points :

L41-43 : the end of this sentence is missing. “are involved “ should be added: “…where many factors…and many others are involved.”

L163: the word “roots” is used twice instead of “shoots and roots”.

Table 1: the first letter of the word “induced” at the end of this table is underlined without any reason.

Author Response

(The authors gave the same response as above.)

Round 2

Reviewer 1 Report

Comments and Suggestions for Authors

While the authors have made commendable efforts in reformulating the article, I still believe it falls short of meeting the standards required for publication in the journal. The topics discussed could benefit from a more nuanced approach, with a clearer authorial perspective and stronger arguments, particularly as this is an opinion manuscript. Furthermore, it would be valuable to include a figure illustrating the role of PEG in inducing abiotic stress, along with a discussion of the associated challenges and benefits

Author Response

Please see our response in the attached file.

Reviewer 3 Report

Comments and Suggestions for Authors

The authors provided substantial improvements of this article. There are much more convincing arguments to support their point of view and bibliographic references are much more extended.

Section 2 is more informative thus interesting and the two added studied cases are relevant.

Only one of my concern was not taken into account: comparing gradual natural stress (drought) to immediate artificial stress (PEG). But I understand that is a different point of view which was not the subject of this article.

Therefore, I find this revised version suitable for publication. I only saw 2 very minor corrections to make, as indicated below.

Minor points :

L116: “-1.48 Mpa” instead of “-1,48 MPa”.

L158: “…, when it was possible.” instead of “…, where it was possible.”

Author Response

Please see our responses in the attached file.
